# DIVERSITY-AWARE CONTINUAL LEARNING WITH LATENT KNOWLEDGE HYPERGRAPH

## ABSTRACT

Continual learning (CL) refers to the ability of models to learn from non-stationary data distribution while transferring and protecting past knowledge. Existing literature in CL has mainly focused on overcoming catastrophic forgetting. However, they often overlook a critical trade-off between parameter efficiency and capacity saturation. Almost all of the existing approaches including architecture-stable and architecture-growing methods struggle to balance parameter efficiency and capacity saturation. This makes them vulnerable to long-term task-incremental CL under storage constraints. In this paper, we propose a novel CL approach, Continual **K**nowledge **HyperG**raph **L**earning (**HyperGKL**), which explicitly addresses the trade-off between parameter efficiency and capacity saturation by efficiently expanding the model's weight space in proportion to the actual capacity increase needed by each new task. Specifically, our approach introduces a unique knowledge hypergraph structure that captures the latent knowledge across tasks and leverages it to measure task diversity and estimate the capacity increase required for each new task. Moreover, we introduce new constraints to ensure parameter efficiency during inference and a fine-grained parameter generator to create task-specific sub-networks that ensure a constant number of trainable parameters over time while accommodating the evolving complexities of tasks. Extensive experiment results show that the proposed approach achieves state-of-the-art results on several benchmark CL datasets, while maintaining low parameter counts.

## 1 INTRODUCTION

Continual Learning (CL) is a pivotal challenge in machine learning, especially in the context of online task-incremental scenarios where new tasks arrive sequentially. Most of the existing CL approaches focus on addressing the issue of *catastrophic forgetting* (Kirkpatrick et al., 2017), i.e., the previously learned tasks are forgotten or negatively impacted after learning new tasks. However, an overlooked issue in existing CL literature is the critical **trade-off between parameter efficiency and capacity saturation**. While *capacity saturation* refers to neural networks becoming inadequate to generalize knowledge from seen tasks as task diversity increases over time, *parameter inefficiency* occurs when networks become over-parameterized and exceed the maximum capacity needed for a good knowledge generalization over seen tasks (inference-time inefficiency) or when the trainable parameters needed for achieving a good performance largely exceeds the memory budget (training-time inefficiency). Existing CL approaches (Li & Zeng, 2023; Gupta et al., 2022), including replay-based, architecture-stable, and expansion-based methods, struggle to strike a balance between these two challenges.

**Contributions:** **(1)** To address the aforementioned issue, we propose a novel CL formulation that *explicitly* considers the trade-off between parameter efficiency and capacity saturation in task-incremental CL scenarios. **(2)** Central to our approach, Continual **K**nowledge **HyperG**raph **L**earning (**HyperGKL**), is the design of a unique knowledge hypergraph structure that captures the latent knowledge across tasks and leverages it to measure task diversity and estimate the capacity increase required for each new task. Specifically, the proposed knowledge hypergraph is a dynamic structure that dissects tasks into multiple skills (latent knowledge) and captures the intricate task interdependencies at a granular level. Each task arriving in the CL setting is decomposed into multiple skills, and each skill is represented by a vertex in the hypergraph. Hyperedges connect vertices that represent skills that are shared by the same task. The knowledge hypergraph is constructed dynamically as the model learns new tasks. For each new task, the model first decomposes the task into multiple skills. It then adds the new skills to the knowledge hypergraph and updates the hyperedges to reflect the

relationships between the new skills and the existing skills. Moreover, the task diversity is assessed by measuring the average shared skills between tasks, where higher values indicate lower diversity. **(3)** To ensure parameter efficiency during inference, the approach introduces new constraints that encourage diversity among skills and facilitates skill-sharing among tasks, offering guidance for capacity expansion while minimizing inefficiencies during inference. At the same time, to overcome the capacity saturation issue, we propose an algorithm that identifies shared skills with the current task, determines the requirement for new skills, and seamlessly integrates skill-specific submodules for network expansion. This approach optimizes the balance between expanding capacity and preserving parameter efficiency. **(4)** Finally, the approach leverages a fine-grained parameter generator through a hypernetwork that dynamically creates task-specific sub-networks to ensure a constant number of trainable parameters over time while accommodating the evolving complexities of tasks, threby enhancing training-time efficiency.

## 2 PROBLEM DEFINITION: DIVERSITY-AWARE PARAMETER-EFFICIENT CL

**Task-incremental Continual Learning (CL):** We focus on the online task-incremental continual learning setting (Li et al., 2019). Consider a sequence of tasks $\mathcal{T}_1, \mathcal{T}_2, ..., \mathcal{T}_T$ arriving to a learner, where each task $\mathcal{T}_t = \{\mathbf{x}_{j,t}, y_{j,t}\}_{j=1}^{n_t}$ consists of $n_t$ training/validation/testing samples from a its own label space. We assume there exist simultaneous ***input distribution shift*** and ***label space drift*** over tasks. Suppose $\boldsymbol{\theta}_t$ denotes the trainable parameters at task $\mathcal{T}_t$ and $\boldsymbol{\Theta}_t = \bigcup_{i=1}^{t} \boldsymbol{\theta}_i$ collects all trainable parameters up to $\mathcal{T}_t$. Let $f(\cdot; \boldsymbol{\Theta}_t)$ denote the neural network at task $\mathcal{T}_t$ parameterized by $\boldsymbol{\Theta}_t$. When a new task $t$ comes, the *main objective* of standard CL is to minimize the task-specific loss on $\mathcal{T}_t$ as well as minimize the forgetting for all seen tasks $\mathcal{T}_1, \mathcal{T}_2, ..., \mathcal{T}_{t-1}$

$$\min_{\boldsymbol{\Theta}_t} \mathcal{L}^{\text{task}}(\boldsymbol{\Theta}_t) + \mathcal{R}^{\text{fgt}}(\boldsymbol{\Theta}_t) = \mathbb{E}_{(\mathbf{x},y) \sim \mathcal{T}_t} l(f(\mathbf{x}; \boldsymbol{\Theta}_t), y) + \mathcal{R}^{\text{fgt}}(\boldsymbol{\Theta}_t), \tag{1}$$

where $l(\cdot, \cdot)$ is the loss function. The challenge of Eq.(1) is to minimize its ***anti-forgetting*** term $\mathcal{R}^{\text{fgt}}(\boldsymbol{\Theta}_t) = \sum_{i=1}^{t-1} \mathbb{E}_{(\mathbf{x},y) \sim \mathcal{T}_i} l(f(\mathbf{x}; \boldsymbol{\Theta}_t), y)$ without the access of $\mathcal{T}_1, \mathcal{T}_2, ..., \mathcal{T}_{t-1}$.

**Motivation:** Recently, due to user preferences or privacy issues, there has been a growing real-world desire for solving CL problems on ***memory-constrained*** local devices. In case of long-term tasks or small memory budget for CL learners, existing CL approaches face the following challenges: (1) Replay-based CL (Rolnick et al., 2019) is sometimes not available as they require extra storage for a subset of previous samples and then replay them; (2) Non-replay based CL, including the *architecture-stable* approaches with a *constant weight space* over time (Von Oswald et al., 2019) and the *expansion-based* approaches that *expands the weight space* by adding parameters/modules over time to accommodate new learned knowledge (Li et al., 2019), although without a replay buffer for data, still has the limitations on parameter efficiency while dealing with the catastrophic forgetting. To summary, while existing approaches mainly focus on dealing with the catastrophic forgetting, one unique *challenge* in the CL literature still remain ***under-explored***: the ***trade-off between Capacity Saturation and Parameter Efficiency***. *Capacity saturation* refers to the *under-parameterization* phenomenon when the parameter space of a neural network is not enough to generalize the knowledge of seen tasks–as the *diversity and complexity of seen tasks* increases over time, the neural network tends to be *saturated* in memorizing seen knowledge, which loses the ability to generalize new knowledge for future tasks. For example, capacity saturation happens when the growing of the neural network size is slower than the growing of task complexity/diversity. *Parameter Efficiency* involves both training-time and inference-time parameter efficiency. While inference-time parameter inefficiency refers to *over-parameterization* phenomenon when the expansion of weight space largely exceed the actual growing of task diversity, training-time parameter inefficiency refers that the size of trainable parameters needed for achieving a good performance largely exceeds the memory budget. Under memory-constrained CL scenarios, both capacity saturation and training/inference-time parameter efficiency are important. Unfortunately, the existing CL approaches cannot achieve a good balance between ensuring parameter efficiency and handling capacity saturation.

Therefore, one unanswered question is that "*how to efficiently expand the weight space to overcome capacity saturation in CL while minimizing the number of parameters in the neural network?*" In order to systematically address this question, we formulate a new CL problem, namely **Di**versity-aware and **P**arameter-efficient CL, which is defined as follows.

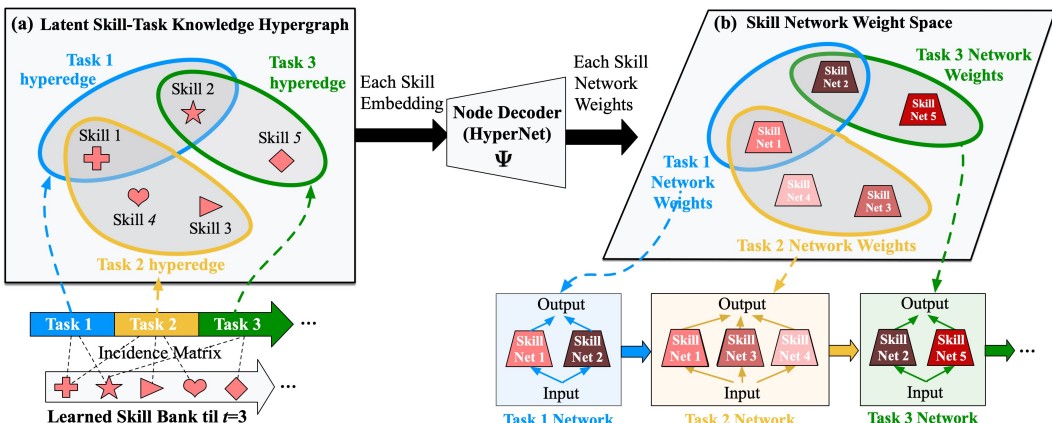

Figure 1: Proposed HyperGKL framework. Different tasks are represented by different colors (blue for task 1, yellow for task 2, and green for task 3). The learned skills–each represents a knowledge type learned from the tasks, are represented in red; different skills are represented by different shapes.

***Definition* 1 (Diversity-aware Parameter-efficient CL).**    On the basis of conventional CL, we explicitly introduce the extra constraints on the parameter efficiency to the CL objective

$$\min_{\mathbf{\Theta}_t} \mathcal{L}^{\text{task}}(\mathbf{\Theta}_t) + \mathcal{R}^{\text{fgt}}(\mathbf{\Theta}_t) + \mathcal{R}_1^{\text{eff}}(\nabla\mathbf{\Theta}_t) + \mathcal{R}_2^{\text{eff}}(\mathbf{\Theta}_t). \tag{2}$$

$\mathcal{R}_1^{\text{eff}}(\nabla\mathbf{\Theta}_t) = c(\nabla\mathbf{\Theta}_t)$ is the training-time parameter efficiency term at task $t$, where $\nabla\mathbf{\Theta}_t$ is the weight gradient at task $t$ and $c(\cdot)$ is a function counting the size of its input gradients or weights. $\mathcal{R}_2^{\text{eff}}(\mathbf{\Theta}_t) = \max(c(\mathbf{\Theta}_t) - c(\mathbf{\Theta}_{t-1}), \gamma\Delta_t(\mathcal{T}_{1:t}))$ is the inference-time parameter efficiency term at task $t$, where $\gamma$ is a scaling hyperparameter. The insight of $\mathcal{R}_2^{\text{eff}}$ is that, in order to achieve a better trade-off between capacity saturation and inference-time parameter efficiency, the *weight space expansion* $c(\mathbf{\Theta}_t) - c(\mathbf{\Theta}_{t-1})$ should be aware of the *diversity gain* $\Delta_t(\mathcal{T}_{1:t})$ when adding the new task $t$ to seen tasks

$$\Delta_t(\mathcal{T}_{1:t}) = g^{\text{diverse}}(\mathcal{T}_{1:t}) - g^{\text{diverse}}(\mathcal{T}_{1:(t-1)}), \tag{3}$$

where $g^{\text{diverse}}(\cdot)$ is a predefined or trainable function that measures the task diversity. The challenge of solving Eq.(2) is *threefold*. **First**, how to define $g^{\text{diverse}}(\cdot)$ to measure the task diversity reasonably? **Second**, how to compute Eq.(3) without the access of $\mathcal{T}_1, \mathcal{T}_2, ..., \mathcal{T}_{t-1}$? **Third**, considering the two parameter-efficiency terms ($\mathcal{R}_1^{\text{eff}}$ and $\mathcal{R}_2^{\text{eff}}$) are not differentiable as $c(\cdot)$ is a non-differentiable function, how to train the model to minimize them with other losses remains a question.

## 3 METHODOLOGY

To solve the Diversity-aware Parameter-efficient CL problem defined in Eq.(2), we propose a new CL learning framework, namely Continual **K**nowledge **HyperG**raph **L**earning (**HyperGKL**). The main idea of HyperGKL is to continuously learn a *dynamic latent knowledge hypergraph* underlying tasks, which consists of (1) trainable *embeddings* of fine-grained knowledge types (namely, skills) that can be shared among seen tasks and (2) *hyperedges* that reveal the complex task relationships and *task diversity* of seen tasks, and then, utilize this hypergraph with a *hypernetwork (HyperNet) based node decoder* to empower a *parameter-efficient* knowledge transfer from previous tasks to future task on the latent knowledge space. An overview of HyperGKL is illustrated in Figure 1.

### 3.1 MODELING TASK DIVERSITY VIA LATENT KNOWLEDGE HYPERGRAPH

To guarantee the balance between inference-time efficiency and capacity saturation ($\mathcal{R}^{\text{eff}}$), it is crucial to modeling and quantify the **task diversity** ($g^{\text{diverse}}$) to determine the minimum capacity increase needed for learning the new tasj's knowledge without catastrophic forgetting on the previous tasks.

**Group-level v.s Parameter-level v.s. Latent Knowledge-level Measurements:**   In recent works out of the parameter-efficient CL context, task diversity has been leveraged in two ***granularity*** levels: (1) *Group level*, where tasks are separated into disjoint groups (Wang et al., 2023) with each group containing tasks having similar distributions, and $g^{\text{diverse}}(\mathcal{T}_{1:t})$ can be the number of seen groups til task $t$; (2) *Parameter/Module level*, where any pair of tasks has partial common weight space

or modules, each parameter/module denotes a piece of information, and $g^{\text{diverse}}(\mathcal{T}_{1:t})$ can be the accumulated number of parameters/modules til task $t$ (Kang et al., 2022; Li et al., 2019). However, both of them have *drawbacks in the parameter-efficient CL* context. Group-level measurement is not accurate as it ignores common knowledge among different groups of tasks and may result in over-estimate of model capacity. Parameter/Module-level measurement requires greedy search in a larger weight space and then apply pruning to cut weights backward, which is not efficient in training time. To tackle these challenges, we propose to learn a ***latent knowledge space*** and then efficiently ***measure the task diversity on the latent space***. On the latent knowledge space, points represent fine-grained knowledge types learned by tasks, which can be shared among different tasks and thus make the task relationships more organized. In this paper, we call these fine-grained ***latent knowledge types*** as ***skills***. Each skill means a type of functional ability trained on a task.

**Latent Knowledge Space:** Suppose $D$ is the dimension of the latent knowledge space. On the latent knowledge space, we assume there exists a set of $M_t$ unique skills seen until task $t$, represented as $\mathcal{S}_t = \{s_j \in \mathbb{R}^D\}_{k=1}^{M_t}$, where $s_j$ is the ***skill embedding*** of the skill type $j$. Each task $i \leq t$ learned a collection of differnt skills, and we denote $\mathcal{C}_i = \{s_j\}_{k \in \mathcal{I}_i} \subseteq \mathcal{S}_t$ as a collection of embeddings of the learned skills on the task $i$, which is a subset of $\mathcal{S}_t$, where $\mathcal{I}_i \subseteq [M_t]$ is a collection of indices indicating the skill identities that comprise the task $i$. For example, as show in Figure 1(a), until $t = 3$ we have $M_3 = 5$, $\mathcal{I}_1 = \{1, 2\}$, $\mathcal{I}_2 = \{1, 3, 4\}$, and $\mathcal{I}_2 = \{2, 5\}$.

*Definition* 2 (**Skill-task Knowledge Hypergraph**). On the latent knowledge space, each seen skill $s_j \in \mathcal{S}_t$ can exist in multiple tasks and each task $i$ may consist of multiple skills. Such skill-task relationships indicate a ***hypergraph*** structure on this latent knowledge space

$$\mathcal{G}_t = (\mathcal{S}_t, \mathcal{E}_t, \mathcal{H}_t), \tag{4}$$

where (1) the fine-grained knowledge types ***(skills) are vertices*** in the hypergraph, whose node features are the trainable skill embeddings $\boldsymbol{S}_t \in \mathbb{R}^{|\mathcal{S}_t| \times D}$; (2) the coarse-grained ***tasks*** $\mathcal{E}_t = [t]$ ***are treated as hyperedges***; and (3) we let $\mathcal{H}_t = \{\mathcal{I}_i\}_{i=1}^t$ denote a collection of skill indicators of all the seen tasks until $t$, which is considered as the connections between skill nodes and represented as a binary-valued ***incidence matrix*** $\boldsymbol{H}_t \in \{0, 1\}^{|\mathcal{S}_t| \times |\mathcal{E}_t|}$. Each entry $\boldsymbol{H}_{t,j,i}$ in the incidence matrix denotes whether the task $i$ learns skill type $j$. The hypergraph $\mathcal{G}_t$ is dynamic and becomes larger and larger as both the number of nodes (skills) and the number of hyperedges (tasks) increase overtime.

**Task Diversity & Skill Disentanglement:** We have modeled fine-grained task relationships on a latent knowledge space, which explicitly indicates the knowledge types shared by every pair of tasks using the incidence matrix. However, to successfully use the hypergraph, there are two additional preassumptions. The first is the knowledge diversity between skills. To encourage the diversity between skills, we introduce a ***negative correlation penalty*** term to the objective

$$\mathcal{L}_{\text{decomp}}(\boldsymbol{S}_t) = \sum_{k \in M_t} -||\boldsymbol{S}_{t,k} - \widehat{\boldsymbol{S}_t}||_2^2 \tag{5}$$

where $\widehat{\boldsymbol{S}_t}$ is the average of all skill embeddings in $\boldsymbol{S}_t$. In addition, since each task is a combination of skills, the knowledge sharing between tasks is not directly but through an organized sharing scheme, that is, the skill indicators. Therefore, in order to encourage the skill-sharing opportunity (i.e., hyperedge overlapping) between tasks to improve the positive knowledge transfer, we introduce an ***association constraint***. Intuitively, the more common skills two tasks share, the more close they are in the embeddings space; and, the more frequent two skills are used in the same task, the closer they are in the skill embedding space. Based on this idea, there is an constraint between $\boldsymbol{S}_t$ and $\boldsymbol{H}_t$ which is inspired by hypergraph Laplacian:

$$\mathcal{L}_{\text{graph}}(\boldsymbol{S}_t, \boldsymbol{H}_t, \boldsymbol{\psi}_{\text{mut}}, \boldsymbol{\psi}_{\text{agg}}) = ||\boldsymbol{H}_t - \widehat{\boldsymbol{H}}||_2^2 \tag{6}$$

where $\widehat{\boldsymbol{H}}_{i,k} = f^{\text{mut}}(\boldsymbol{s}_k, \boldsymbol{e}_i; \boldsymbol{\psi}_{\text{mut}})$ and $\boldsymbol{e}_i = f^{\text{agg}}(\{\boldsymbol{s}_k | k \in [M_t], \boldsymbol{H}_{t,k,i} = 1\}; \boldsymbol{\psi}_{\text{agg}})$. $f^{\text{agg}}(\cdot; \boldsymbol{\psi}_{\text{agg}})$ is the skill aggregation function parameterized by $\boldsymbol{\psi}_{\text{agg}}$, which combines multiple skills and reconstructs the task, and $f^{\text{mut}}(\cdot, \cdot; \boldsymbol{\psi}_{\text{mut}})$ is the measurement computing the association score of each skill $k$ in each task $i$. Here, $\boldsymbol{\psi}_{\text{agg}}, \boldsymbol{\psi}_{\text{mut}}$, The final interence-time parameter effceny loss is

$$\mathcal{R}_2^{\text{eff}}(\boldsymbol{\Theta}_t, \boldsymbol{S}_t, \boldsymbol{H}_t, \boldsymbol{\psi}_{\text{mut}}, \boldsymbol{\psi}_{\text{agg}}) = \mathcal{L}_{\text{graph}} + \mathcal{L}_{\text{decomp}} + \max(c(\boldsymbol{\Theta}_t \setminus \boldsymbol{\Theta}_{t-1}), \gamma|\mathcal{C}_t \setminus \mathcal{S}_{t-1}|). \tag{7}$$

With the knowledge hypergraph with diversified skill embeddings, we can define the task diversity function as follows: given a subset of tasks $\mathcal{T}_\mathcal{U} \subseteq [t]$, their diversity can be calculated as the number of diversified knowledge types (skills) learned by tasks $g^{\text{diverse}}(\mathcal{T}_\mathcal{U}; \mathcal{G}_t) = |\bigcup_{i \in \mathcal{T}_\mathcal{U}} \mathcal{C}_i| = |\bigcup_{i \in \mathcal{T}_\mathcal{U}} \mathcal{I}_i|$.

## 3.2 Knowledge Hypergraph Evolution for Inference-time Efficiency

**Granularity of Skill Submodule:** We let each skill $j$ be associated with a small trainable neural network $\phi_j$, a *skill-specific subnetwork*, embedded in the full task neural network. One straightforward way of solving the last term of Eq.(7) is to let the network growing be simply guided by $\mathcal{C}_t \setminus \mathcal{S}_{t-1}$. That is, we introduce new parameters as $\mathbf{\Theta}_t \setminus \mathbf{\Theta}_{t-1} = \boldsymbol{\theta}_t = \{\phi_{j'}\}_{j' \in \mathcal{C}_t \setminus \mathcal{S}_{t-1}}$ and then train the parameters. However, the topology and number of parameters of each skill-specific network actually impact the capacity gain after adding each skill. This can be treated as a hyperparameter, which is the $\gamma = c(\phi_{j'})$ in Eq.(2). For simplicity, we use the same architecture for each skill-specific network. Specifically, we use a autoencoder-based low-rank network for each skill. Therefore, we can rewrite: $\max(c(\mathbf{\Theta}_t \setminus \mathbf{\Theta}_{t-1}), \gamma | \mathcal{C}_t \setminus \mathcal{S}_{t-1} |) = | \mathcal{C}_t \setminus \mathcal{S}_{t-1} |$.

**Novel Knowledge Discovery & Hypergraph Growth:** When leveraging network growth that expands the weight space to mitigate capacity saturation and forgetting, we anticipate that the network growth should be as small as possible, that is, minimizing inference-time parameter efficiency. We resort to optimizing Eq.(7) to achieve this goal. One remaining challenge is that the third term of Eq.(7) is non-differentiable and forbidding of using gradient descent. To handle this challenge, we proposed the following searching steps to find the *optimal* new set of skills $\mathcal{C}_t \setminus \mathcal{S}_{t-1}$ for every new task: 1) identifying any existing skills shared with the current task; 2) identifying the number of new skills needed by the current task; 3) adding modules for new skills and train the model. Specifically, to minimize $|\mathcal{C}_t \setminus \mathcal{S}_{t-1}|$, we split $\mathcal{C}_t = \{\mathcal{C}_t^{pre}, \mathcal{C}_t^{new}\}$ into two parts, where $\mathcal{C}_t^{pre} \subset \mathcal{S}_{t-1}$ is seen in previous tasks and $\mathcal{C}_t^{new}$ is the new skills needed by the current task, and we aim to solve the two parts in separate steps. Given the previously trained skill embeddings $\boldsymbol{S}_{t-1} \in \mathbb{R}^{|\mathcal{S}_{t-1}| \times D}$ and the associated skill submodules $\mathbf{\Theta}_{t-1} = \{\phi_j | j \in |\boldsymbol{S}_{t-1}|\}$, we first retrieve at most $k$ skill submodules that give the best prediction results, where $k$ is an upperbound hyperparameter. We use the validation set of $\mathcal{T}_t$ and compute $f(x; \phi_j)$ to obtain the validation errors using each skill submodule $\phi_j$. Then, we jointly train the top-$k$ submodules and compute the validation error, which is compared with an error threshold $\epsilon$. If the existing skills do not achieve a satisfied validation error, we then introduce new skills. In order to determine how many new skills are need by the current task, we employ evolutionary search with a search space of $k$.

## 3.3 HyperNet-based Node Decoder for Training-time Efficiency

While attempting to minimize the training-time efficiency, i.e. $\mathcal{R}_1^{\text{eff}}(\nabla \mathbf{\Theta}_t)$ in Eq.(2), we consider two reasons that may lead to the inefficiency during the training time: (1) the number of skill-specific subnetworks increases over time as the number of tasks increases, and the strategy searching for new skills may revisit these previously learned skill networks. Storing a large number of skill-specific networks is forbidden with memory limits. (2) When the number of maximum skills $k$ per task is large, the trainable parameters at each task tend to exceed the memory budget. To overcome these challenges, we can also borrow the benefit of the latent knowledge hypergraph learned in the previous sections. We propose to operate the knowledge transfer from previous tasks to the current task on the latent knowledge space, instead of on the original task network's weight space. To achieve this, we leverage a HyperNet-based node decoder to bridge the gap between the latent knowledge space and the task networks' weight space.

**Node Decoder on Hypergraph (Parallel Parameter Generation Trick)**: Assuming a task network $\boldsymbol{\theta}_t = \{\boldsymbol{\theta}_t^{\text{base}}, \boldsymbol{\theta}_t^{\text{adpt}}\}$ consists of base modules $\boldsymbol{\theta}_t^{\text{base}}$ shared by all tasks and task-adaptive modules $\boldsymbol{\theta}_t^{\text{adpt}} = \{\phi_{t,j} | j \in I_t\}$, where each adaptive module consists of multiple submodules, each of which is responsible to learn a skill type $j$ on the task. We leverage a HyperNet-based **node decoder** applied to each skill node in a task-specific hyperedge in the hypergraph $\mathcal{G}_t$. For each previous task $i$, with its skill set information $I_t$ encoded in the incidence matrix, the adaptive module of task $t$ can be generated through a parallel parameter generation trick

$$\boldsymbol{\theta}_t^{\text{adpt}} = \{h(\boldsymbol{s}_j; \mathbf{\Psi}) | j \in I_t\} \tag{8}$$

where $h(\cdot; \mathbf{\Psi}) : \mathbb{R}^D \to \mathbb{R}^F$ is a HyperNet shared by all skills (all nodes), which takes as input a skill embedding and generates the weights of the subnetwork corresponding to the input skill. $\mathbf{\Psi}$ denotes trainable parameters and $F$ is the dimension of the weight space of each skill subnetwork. As shown in Figure 1, the HyerNet is used as a node decoder on the knowledge hypergraph, and therefore, can

be trained in an end-to-end manner using the loss from the task. The forward function of the task network is $f(x; \boldsymbol{\theta}_t^{\text{base}}, \boldsymbol{\theta}_t^{\text{adpt}}) = \frac{1}{|I_i|} \sum_{j \in I_t} f(x; \boldsymbol{\theta}_t^{\text{base}}, h(\boldsymbol{s}_j; \boldsymbol{\Psi}))$.

**Training-time Efficiency**. While the weight space of the generator $\boldsymbol{\Psi}$ is fixed over time, which is efficient in parameter consumption, the task networj's weight space $\boldsymbol{\theta}_t^{\text{adpt}} = \{\boldsymbol{\phi}_{i,j} | j \in I_t\}$ generated by $\boldsymbol{\Psi}$ is dynamic and corresponds to the capacity needs. As a result, although the weight space of the task network expands over time according to the needed capacity increase, the number of trainable weights remain constant over time, that is, $\mathcal{R}_2^{\text{eff}}(\nabla \boldsymbol{\Theta}_t^{\text{adpt}}) = a(\boldsymbol{\Psi})$. The reason of using the hypernetwork to improve the training-time efficiency is that the number of seen skills usually grow up to hundreds or thousands in long-term task-incremental CL. Directly learning and discovering previous skills that are useful to the current task requires simultaneously training a large number of skill modules in the early stage, which is not quite efficient. In additional, the task networks can be generated by passing multiple skills through the node-level HyperNet in parallel in a batch.

## 4 EXPERIMENTS

**Datasets:** We use *four* popular continual learning datasets. (1) **Permuted MNIST** consists of 10 tasks, where each task is a variant of MNIST (LeCun, 1998) after applying a task-personalized deterministic permutation to the input image pixels of all input images. (2) **Omniglot Rotation** is composed of 100 tasks constructed from the raw Omniglot (Yoon et al., 2019) having more than 1200 classes. Each task has 12 distinct classes, which is generated from the raw images by adding their rotation version of 90, 180, and 270 degrees. (3) **5-Datasets** is a mixture of 5 different vision datasets (Saha et al., 2021), including MNIST (task 1), CIFAR-10 (Krizhevsky et al., 2009) (task 2), SVHN (Netzer et al., 2011) (task 3), FashionMNIST (Xiao et al., 2017) (task 4), and notMNIST (Bulatov, 2011) (task 5). (4) **CIFAR100-10** is constructed by dividing the 100 classes of CIFAR-100 (Krizhevsky et al., 2009) into 10 tasks with 10 distinct classes per task. Task samples do not repeat over tasks. In Omniglot Rotation, 5-Datasets, and CIFAR100-10, tasks has their unique label spaces.

**Baselines:** We compared our method with *two families* of *non-rehearsal-based* CL baselines. (1) *Regularization-based* approaches, where the model architecture remains fixed over tasks and the catastrophic forgetting problem is handled by regularization techniques, include **Naive FINETUNE** (a naive sequential training strategy where a single model is trained continually on sequentially coming tasks), **EWC** (Kirkpatrick et al., 2017), **HAT** (Serra et al., 2018), **GPM** (Saha et al., 2020), and the HyperNet-based method **HCL** (von Oswald et al., 2020; Hemati et al., 2023); (2) *Expansion-based* approaches, where the model architecture and the parameter space dynamically expands and changes over time, include **Independent** (a naive strategy where each task learns a new model from scratch without using the previously learned models), **Learn to Grow** (Li et al., 2019), **PAR** (Wang et al., 2023) (grouping tasks and different tasks either share all weights or share no weight), **SupSup** (Wortsman et al., 2020) (finding supermasks within a randomly initialized network for each task), and **WSN** (Kang et al., 2022) (jointly training weights and find supermasks for each task). *Moreover*, we add "Multitask Learning with SparseMoE (**MTL-MoE**)" baseline, where all the tasks are learned simultaneously in a SparseMoE model (Gupta et al., 2022). MTL-MoE is not a CL approach but will serve as upper bound on average accuracy on all tasks.

**Model Architectures & Hyperparameters:** We use a multi-head configuration for all experiments, where each task train their own classification *head* function. For the *backbone*, to demonstrate fair comparisons between different methods, we encourage all the methods to start with the same inference-time model architecture at the *first* task. Consider that several model layers in the proposed HyperGKL consist of multiple parallel subnetworks (i.e., skill submodules), the baselines also follow such multi-subnetwork paralleled structures. For regularization-based baselines and Independent, we use two-layered MLP with $k$ parallel subnetworks per layer and 100 neurons per subnetwork for Permutated MNIST, use LeNet (Al-Jawfi, 2009) with $k$ parallel subnetworks at the second CNN layer for Omniglot Rotation, use AlexNet (Serra et al., 2018) with $k$ parallel subnetworks at the second CNN layer and the first fully-connected layer for CIFAR100-10, and use ResNet18 (Kang et al., 2022) with $k$ parallel subnetworks at the first CNN layers in block 1 & 3 for 5-Datasets. PAR and Learn to Grow begin with $k$ parallel subnetworks per layer and then expand the number of subnetworks using their expansion algorithms. For SupSup, WSN, and MTL-MoE, we use a supernetwork having similar model structures except that $\rho T k$ parallel subnetworks at each compositional layer. While each task in SupSup and WSN learns a sparse task-adaptive parameter allocation mask for its inference model, MTL-MoE learns an additional gating function to route each task to $k$ subnetworks.

Table 1: Comparison of performance and resource utilization between different methods. For model architectures, we use $\rho = 0.6$, $k = 4$, $D = 8$ for "Permuted MNIST" and $\rho = 0.3$, $k = 4$, $D = 20$ for "Omniglot Rotation". **1**: Non-CL baseline (as upper bound). **2**: Regularization-based CL approaches. **3**: Expansion-based CL approaches. **N/A**: Not Applicable. TPS and IPS values are shown in millions.

| # | Method | Permuted MNIST (10 tasks) | | | | Omniglot Rotation (100 tasks) | | | |
|---|---|---|---|---|---|---|---|---|---|
| | | ACC↑ | BWT↑ | TPS↓ | IPS↓ | ACC↑ | BWT↑ | TPS↓ | IPS↓ |
| **1** | **MTL-MoE** | 0.978 | N/A | 2.13 | 0.35 | 0.884 | N/A | 48.81 | 25.45 |
| | **Naive FINETUNE** | 0.782 | -0.210 | 0.35 | 0.35 | 0.614 | -0.424 | 25.45 | 25.45 |
| | **EWC** | 0.920 | -0.031 | 0.35 | 0.35 | 0.714 | -0.114 | 25.45 | 25.45 |
| **2** | **GPM** | 0.944 | -0.024 | **0.35** | 0.35 | 0.834 | -0.032 | **25.45** | 25.45 |
| | **HCL** | 0.949 | -0.022 | 3.19 | 0.35 | 0.842 | -0.010 | 41.23 | 25.45 |
| | **Independent** | 0.783 | -0.261 | 0.35 | 0.35 | 0.493 | -0.502 | 25.45 | 25.45 |
| | **Learn to Grow** | 0.918 | -0.004 | 3.55 | 2.13 | 0.820 | -0.007 | 44.71 | 48.81 |
| **3** | **PAR** | 0.958 | -0.013 | 0.35 | 0.35 | 0.805 | -0.028 | 25.45 | 25.45 |
| | **SupSup** | 0.963 | **0** | 3.55 | 2.13 | 0.581 | **0** | 44.71 | 48.81 |
| | **WSN** | 0.964 | **0** | 2.13 | 0.35 | 0.856 | **0** | *48.81* | 25.45 |
| | **Ours** | **0.966** | -0.004 | 0.99 | **0.35** | **0.864** | -0.008 | 28.53 | **25.45** |

**Evaluation Metrics:** We evaluate our approach against baselines with respect to both the overall performance and model efficiency. (1) First, following (Kang et al., 2022), we measure the *overall performance* of CL approaches using two metrics: **Average Accuracy (ACC)** and **Backward Transfer (BWT)**. Suppose $A_{i,j}$ denotes the test accuracy for task $j$ after training on task $i$. ACC measures the average of the classification accuracy on all tasks using the final model: ACC $= \frac{1}{T}\sum_{j=1}^{T} A_{T,j}$. BWT measures the average forgetting on past tasks: BWT $= \frac{1}{T-1}\sum_{j=1}^{T-1} A_{T,j} - A_{j,j}$. Negative BWT means forgetting. (2) Second, we use two metrics to measure the *overall model efficiency* of CL approaches to show their resource utilization: **Trainable Parameter Size (TPS)** and **Inference-time Parameter Size (IPS).** Suppose $S_i^{\text{train}}$ denotes the number of *trainable* parameters on task $i$ and $S_i^{\text{infer}}$ denotes the number of all parameters in the inference model on task $i$. TPS is the maximum number of trainable parameters over all tasks: TPS $= \max_{i=1}^{T} S_i^{\text{train}}$. IPS is the number of parameters for the largest inference models over all tasks: IPS $= \max_{i=1}^{T} S_i^{\text{infer}}$. For fair comparison, we encourage all comparative methods to initialize with a same-sized inference model at the first task.

Details on datasets, implementation, and hyperparameters are provided in Appendix.

## 4.1 MAIN RESULTS

We report the results after training the final task in Table 1, Table 2 (in appendix), and Figure 2, where the proposed method is compared against all baselines on 4 datasets. All experiments run on a single-GPU of NVIDIA A100. Each experiment repeatedly run 5 times with different random seeds.

**Performance Comparison.** In general, our proposed approaches demonstrated superior ACC and BWT performance rather than baselines. In particular, we observe that the methods using adaptive partial weight sharing (WSN and our method) typically outperformed the full-weight-sharing and zero-weight-sharing baselines with a large margin, which demonstrates that adaptive partial weight sharing is crucial in overcoming forgetting in CL problems. In addition, the proposed HyperGKL outperformed WSN in ACC while slightly tradeoff the BWT under the same experimental setting. Such a BWT is relatively small in comparison with other effective baselines and might be due to the skill-level HyperNet retraining over tasks. Yet we obtain a better ACC performance as WSN baseline sometimes suffered from structure disruption due to model sparsity.

**Resource Utilization Comparison.** To create the same inference models, CL approaches based on partial weight sharing (including our approach, SupSup and WSN) typically requires *larger* TPS than the full/zero-weight-sharing ones (except HCL). The extra parameters are needed to discover the correct weight subspace shared among tasks for better performance. This can be observed from the TPS and IPS results in Table 1 and Table 2. While a larger TPS is usually necessary for better performance, among the partial-weight-sharing methods, our approach demonstrated the smallest

TPS, which shows the best model efficiency. We successfully leveraged the fine-grained HyperNet to save the computation resource during the discovery of inter-task shareable weight subspaces.

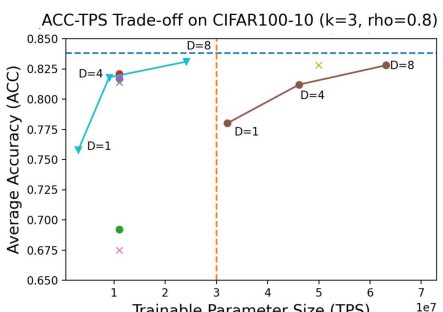 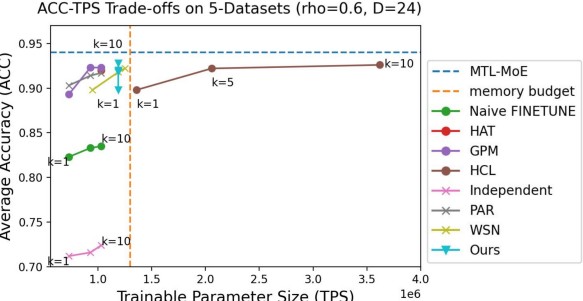

(a) Impact of task/skill embedding size $D$          (b)  Impact of maximum skill number per task $k$

Figure 2: Comparison of Performance-Efficiency Trade-offs (a) on "CIFAR100-10" consisting of 10 tasks, where we show the impact of the embedding dimension $D = 1, 4, 8$ of HyperNet; and (b) on "5-Datasets" with 5 tasks, showing the impact of the maximum skill number per task $k = 1, 5, 10$.

**Comparison of Performance-Efficiency Trade-offs.** To provide a more comprehensive evaluation of CL methods, we illustrate their performance-efficiency trade-offs in Figure 2, where the x-axis shows the TPS and the y-axis shows the ACC. For fair comparison, all methods within the same figure, if using the same $D$ or $k$, start with a same-sized inference model at the *first* task. Methods on the top-left corner have better trade-offs: higher ACC with smaller TPS. Compared with full/zero-weight-sharing methods (PAR, Independent, and regularization-based methods except HCL), our proposed partial-weight-sharing method achieves higher ACC with nearly the same TPS under the memory budget. The HyperNet-based CL method (HCL) requires a very high TPS for coarse-grained weight space learning, which significantly exceeds the memory budget and might suffer from both capacity saturation (if task architecture is too small) and overfitting at early tasks (if task network is too large). In contrast, our method learns finer-grained weight spaces with latent knowledge relationship discovery, which reduces the two limitations. Other partial-weight-sharing methods (SupSup and WSN) continuously expand the weight space and thus require large TPS; instead, our method leverage a fix-sized HyperNet to disentangle the dynamic-sized weight space.

**Visualization of Learned Knowledge Hypergraph.** Figure 3 visualizes the incidence matrix of the learned knowledge hypergraph on CIFAR100-10 among 10 tasks. Each row $i$ of an incidence matrix indicate the task $\mathcal{T}_i$; each column $j$ indicates the discovered skill with index $j$. Entries are binary; the entry $(i, j)$ equals to one (denoted as green squares) denotes that the skill $j$ is discovered and learned at task $i$. We consider the same inference model with $k = 3$ and compare the weight sharing strategies pre-defined by Independent (Figure 3(d)), pre-defined by regularization-based methods (Figure 3(e)), learned by PAR via task grouping (Figure 3(f)), and learned by the proposed method (Figure 3(a-b)). SupSup and WSN do not learn the disentangled knowledge/skills and thus do not participate in the visualization. While Figure 3(d) shows none weight sharing among tasks, Figure 3(e) shows complete/full weight sharing among tasks, and Figure 3(f) shows either full or none weight sharing among tasks, Figure 3(a-b) demonstrate the partial weight sharing strategy learned by our method, which achieved the best ACC performance. Also, while baselines doe not explicitly demonstrate how much knowledge is transferred among tasks, our CL approach leverages the latent hypergraph that explicitly provides the *interpretability* of knowledge transfer.

## 4.2 ABLATION STUDIES

**Impacts of Task/Skill Embedding Size** ($D$).    Figure 2(a) illustrates the influence of the input embedding dimension of HyperNet. In general, for both HCL and the proposed method, larger $D$ results in better ACC performance and larger TPS. With the same $D$, the TPS of our method is constantly smaller than that of HCL. This is because the HCL generates the entire weight space (with HyperNet complexity $O(kD)$), while our method generates the weight subspace for each of fine-grained knowledge types (with HyperNet complexity $O(D)$). We also observe that given larger $D >= 4$, our approach achieved better ACC performance than HCL. The larger $D$ implies higher potential to represent knowledge disentanglement in our approach and higher potential to represent task difference in HCL. However, HCL is limited to a fixed weight space and thus cannot obtain the best model capacity that fit each task's complexity, especially when task complexity shifts dynamically. In contrast, our method can generate capacity-aware networks with different widths.

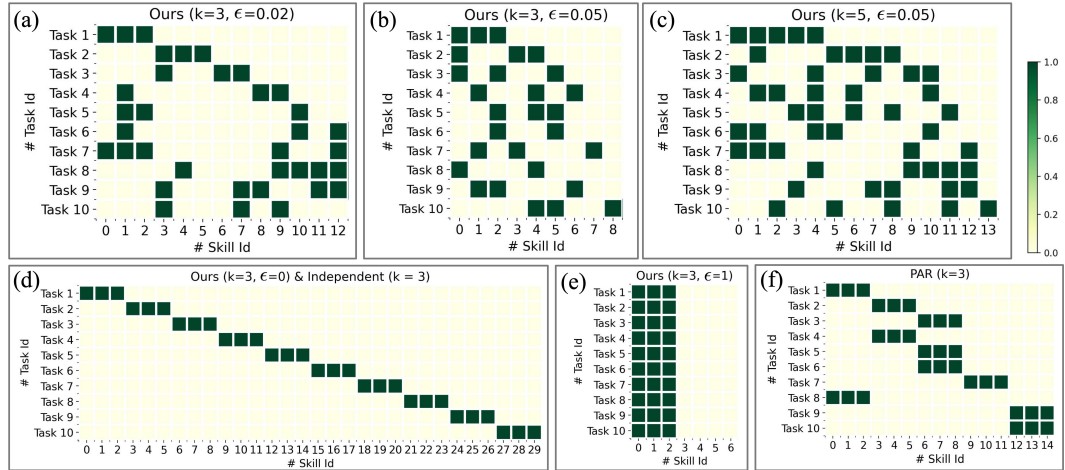

Figure 3: Visualization of the incidence matrix of the latent knowledge hypergraph learned by our proposed method on CIFAR100-10. The performance: (a) ACC=0.830, (b) ACC=0.831, (c) ACC=0.834, (d) ACC=0.671, and (e) ACC=0.692. Among them, (d) and (e) are edge cases. (d) and (e) also demonstrates the weight sharing of Independent and regularization-based approaches, respectively. (f) shows the learned weight sharing scheme of PAR with ACC=0.816 performance.

Moreover, since our knowledge transfer happens on the fine-grained implicit skill level, our partial-weight-sharing strategy can potentially avoid negative knowledge transfer, yet the full-weight-sharing strategy of HCL may transfer conflicting knowledge from task to task.

**Impacts of Maximum Skill Number per Task** ($k$). Figure 2(b) illustrates the impact of the maximum number of skills per task. In general, given the same $k$ value, our approach achieved better ACC performance than baselines. Aor all approaches, the experiment with larger $k$ ($k \leq 10$) obtains higher ACC. Basically, we can increase $k$ to get a better model. However, TPS of all baselines would be traded off for larger $k$ and higher ACC. In contrast, the TPS of our method is nearly constant for different $k$, which suggests that our method is more resource-friendly during the training time rather than baselines. Given a memory budget in the real-world application, there is upper bounds for $k$ and ACC for baselines, while our method can use a larger $k$ than the baseline's upper bound.

**Impacts of Evolution Plasticity** ($\epsilon$). Figure 3(a), (b), (d), and (e) shows the impact of $\epsilon$ on the latent knowledge hypergraph and the overall ACC performance, while fixing $k = 3$. The larger the $\epsilon$, the more skills shared among tasks and there is more chance for weight sharing among different tasks' networks. A special case $\epsilon = 1$ shown in Figure 3(e) indicates full weight sharing. The performance of Figure 3(a)(b) is significantly better than Figure 3(e) because an over-large $\epsilon$ might results in negative transfer. Inversely, the smaller $\epsilon$, the less skills shared among tasks. However, an over-small $\epsilon$ might results in insufficient positive transfer, meaning that some useful knowledge in previous tasks is not utilized by future tasks. A special case $\epsilon = 0$ shown in Figure 3(d) indicates none weight sharing, whose performance is significantly lower than Figure 3(a)(b). Therefore, it is crucial to search a middle-valued $\epsilon$ that can obtain an optimal knowledge hypergraph among tasks. In practice, we searched $\epsilon \in [0, 1]$ and found the best $\epsilon = 0.05$ that gives the best ACC performance.

## 5 CONCLUSIONS

In this work, we have addressed the challenging problem of task-incremental CL by introducing a novel framework that focuses on the trade-off between parameter efficiency and capacity saturation. Our approach recognizes that the existing CL methods struggle to balance these two crucial aspects, leading to vulnerabilities in long-term task-incremental settings with limited memory resources. To overcome this issue, we have introduced the concept of diversity-aware and parameter-efficient CL, where we leverage a unique knowledge hypergraph structure to capture task diversity and estimate the required capacity increase for each new task. Moreover, we have introduced techniques for optimizing the hypergraph growth and ensuring parameter efficiency through fine-grained parameter generation using a fixed-sized hypernetwork. Overall, our framework develops a robust CL solution that can adapt to the evolving complexities of tasks while minimizing the number of trainable parameters.

The related work is provided in the Appendix.

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

# A  APPENDIX

## A.1  EXPERIMENTAL SETUP DETAILS

**Datasets:** We use *four* popular continual learning datasets. (1) **Permuted MNIST** consists of 10 tasks, where each task is a variant of MNIST (LeCun, 1998) after applying a task-personalized deterministic permutation to the input image pixels of all input images. Different tasks share the same label space. (2) **Omniglot Rotation** is composed of 100 tasks, which is constructed from raw Omniglot (Yoon et al., 2019) that has more than 1200 classes. Each task has 720/240 train/test samples from distinct 12 classes, which is generated from the raw images by adding their rotated version in 90, 180, and 270 degrees. (3) **CIFAR100-10** is constructed by randomly dividing the 100 classes of an image classification dataset CIFAR-100 (Krizhevsky et al., 2009) into 10 tasks, with 10 distinct classes per task. (4) **5-Datasets** is a mixture of 5 different image classification datasets (Saha et al., 2021), including MNIST (LeCun, 1998) (task 1), CIFAR-10 (Krizhevsky et al., 2009) (task 2), SVHN (Netzer et al., 2011) (task 3), FashionMNIST (Xiao et al., 2017) (task 4), and notMNIST (Bulatov, 2011) (task 5). In 5-Datasets, Omniglot Rotation, and CIFAR100-10, each task has its own distinct label space and there is no overlapping between task samples.

**Baselines:** We compared our method with *two families* of *non-rehearsal-based* CL baselines. (1) *Regularization-based* approaches, where the model architecture remains fixed over tasks and the catastrophic forgetting problem is handled by regularization techniques, include **Naive FINETUNE** (a naive sequential training strategy where a single model is trained continually on sequentially coming tasks), **EWC** (Kirkpatrick et al., 2017), **HAT** (Serra et al., 2018), **GPM** (Saha et al., 2020), and the HyperNet-based method **HCL** (von Oswald et al., 2020; Hemati et al., 2023); (2) *Expansion-based* approaches, where the model architecture and the parameter space dynamically expands and changes over time, include **Independent** (a naive strategy where each task learns a new model from scratch without using the previously learned models), **Learn to Grow** (Li et al., 2019), **PAR** (Wang et al., 2023) (grouping tasks and different tasks either share all weights or share no weight), **SupSup** (Wortsman et al., 2020) (finding supermasks within a randomly initialized network for each task), and **WSN** (Kang et al., 2022) (jointly training weights and find supermasks for each task). *Moreover*, we add "Multitask Learning with SparseMoE (**MTL-MoE**)" baseline, where all the tasks are learned simultaneously in a SparseMoE model (Gupta et al., 2022). MTL-MoE is not a CL approach but will serve as upper bound on average accuracy on all tasks.

**Model Architectures:**  We consider a multi-head configuration for all experiments in the paper, where each task train their own classification head function. As for the backbone layer, in order to demonstrate fair comparisons between different methods on the performance-efficiency trade-offs, we encourage all the baselines to have the same-sized inference-time model architecture. Since the model layers in the proposed HyperGKL consist of multiple parallel subnetworks (i.e., skill submodules), the model architectures of baselines follow such multi-subnetwork paralleled structure as well. For *regularization-based CL* approaches and Independent, we use two-layered MLP with $k$ parallel subnetworks per layer and 100 neurons per subnetwork for Permutated MNIST, use LeNet (Al-Jawfi, 2009) with $k$ parallel subnetworks at the second CNN layer for Omniglot Rotation, use AlexNet (Serra et al., 2018) with $k$ parallel subnetworks at the second CNN layer and the first fully-connected layer for CIFAR100-10, and use ResNet18 (Kang et al., 2022) with $k$ parallel subnetworks at the first CNN layers in block 1 and block 3 for 5-Datasets. For PAR and Learn to Grow, we let each task or each group's expert model has the same architectures as above; those baselines expand the number of experts using their own algorithms. For SupSup and WSN, we use similar model structures except that $\rho T k$ parallel subnetworks at each compositional layer; given this model as the supernetwork, those baselines learn the task-adaptive parameter allocation for each tasj's inference model. $1/T \leq \rho \leq 1$ is a hyperparameter that determines the pre-defined maximum number of skills in $T$ tasks. For MTL-MoE, we use similar model structures with $\rho T k$ parallel subnetworks at each compositional layer; in addition, we use an additional gating function to route each task to $k$ experts; in this way, the inference model of each task also consists of $k$ subnetworks. The HyperNets in both HCL and our method are three-layered MLP with $D$-dimensional input, $D$ neurons per layer, and multi-head output spaces where each output space is the weight space of a compositional layer or its submodule in the main network.

Table 2: Comparison of performance and resource utilization between different methods. For model architectures, we use $\rho = 0.7$, $k = 3$, $D = 8$ for "CIFAR100-10" and use $\rho = 0.6$, $k = 10$, $D = 24$ for "Five Dataset". **1**: Non-CL baseline (as upper bound). **2**: Regularization-based CL approaches. **3**: Expansion-based CL approaches. **N/A**: Not Applicable. TPS and IPS values are shown in millions.

| # | Method | CIFAR100-10 (10 tasks) | | | | 5-Datasets (5 tasks) | | | |
|---|--------|------|------|------|------|------|------|------|------|
| | | ACC↑ | BWT↑ | TPS↓ | IPS↓ | ACC↑ | BWT↑ | TPS↓ | IPS↓ |
| 1 | **MTL-MoE** | 0.838 | N/A | 49.99 | 11.05 | 0.932 | N/A | 1.25 | 1.03 |
| | **Naive FINEUNE** | 0.692 | -0.393 | 11.05 | 11.05 | 0.835 | -0.226 | 1.03 | 1.03 |
| | **EWC** | 0.782 | -0.125 | 11.05 | 11.05 | 0.898 | -0.096 | 1.03 | 1.03 |
| **2** | **HAT** | 0.821 | -0.104 | 11.05 | 11.05 | 0.920 | -0.053 | 1.03 | 1.03 |
| | **GPM** | 0.817 | -0.071 | **11.05** | 11.05 | 0.923 | -0.047 | **1.03** | 1.03 |
| | **HCL** | 0.828 | -0.067 | 63.14 | 11.05 | 0.926 | -0.045 | 3.62 | 1.03 |
| | **Independent** | 0.675 | -0.476 | 11.05 | 11.05 | 0.724 | -0.430 | 1.03 | 1.03 |
| | **Learn to Grow** | 0.808 | -0.021 | 69.67 | 49.99 | 0.908 | -0.012 | 1.46 | 1.25 |
| **3** | **PAR** | 0.814 | -0.042 | 11.05 | 11.05 | 0.917 | -0.040 | 1.03 | 1.03 |
| | **SupSup** | 0.782 | **0** | 69.67 | 49.99 | 0.799 | **0** | 1.46 | 1.25 |
| | **WSN ($c$=0.3)** | 0.828 | **0** | 49.99 | 11.05 | 0.922 | **0** | 1.25 | 1.03 |
| | **Ours** | **0.831** | -0.011 | 24.14 | 11.05 | **0.928** | -0.017 | 1.19 | 1.03 |

**Hyperparameters:** The task/skill embedding size is set to $D = 8$ for Permutted MNIST and CIFAR100-10, set to $D = 20$ for Omniglot Rotation, and set to $D = 24$ for 5-Datasets.

# B RELATED WORK

**Replay based Lifelong learning** has emerged as a fundamental strategy to mitigate catastrophic forgetting in continual learning (CL) settings. It draws inspiration from several key techniques in CL literature. **(1)** Experience replay (Mnih et al., 2016), involves storing and randomly sampling past experiences to revisit and learn from them, effectively reducing the impact of forgetting. **(2)** Generative replay (Shin et al., 2017), leverages generative models to recreate past data, allowing the model to train on a combination of new and old data, thus aiding in retaining previous knowledge. **(3)** Pseudo-rehearsal (Robins, 1995), involves rehearsal of previous tasks using a network's current knowledge to prevent catastrophic forgetting. **(4)** Episodic memory consolidation (Lopez-Paz & Ranzato, 2017), allows models to consolidate and selectively replay important past experiences. These replay-based methods collectively contribute to the foundation of lifelong learning by addressing the crucial challenge of retaining knowledge across tasks without overfitting to the past, making them valuable techniques in the field of continual learning.

**Regularization-based Methods** constitute another pivotal category within the continuum of CL strategies. These techniques are motivated from the rich landscape of regularization approaches in machine learning. Elastic Weight Consolidation (EWC) (Kirkpatrick et al., 2017), was among the pioneering works in this research direction. EWC regularizes the neural network's weights to protect previously learned parameters while adapting to new tasks. Further advancing the regularization-based paradigm, the Progress and Compress (P and C) algorithm (Schwarz et al., 2018) combines elastic weight consolidation with techniques to compress the model, reducing its computational demands. Moreover, synaptic intelligence (Zenke et al., 2017), introduces regularization terms that adaptively constrain the neural network's weights based on the importance of each parameter. These regularization-based CL methods have demonstrated their efficacy in preserving past knowledge by constraining the updates on parameters that are critical for earlier tasks while allowing adaptation to new information, thereby facilitating the pursuit of continual learning objectives.

**Expansion-based Methods** represent a prominent category of approaches in the field CL. These methods address the challenge of learning new tasks by dynamically expanding the neural network's architecture to accommodate increasing knowledge. A major contribution to this category is the Progressive Neural Network (Rusu et al., 2016). PNN incrementally adds new neural network modules for each task, allowing the model to expand its capacity with task complexity. Another approach is the

Memory Aware Synapses method (Aljundi et al., 2018), which leverages a separate memory matrix to store task-specific information, enabling efficient access to past knowledge. Furthermore, (Joseph et al., 2021) presented the strategy consolidation network, which expands the network by adding new neural modules while retaining shared knowledge. Expansion-based methods have demonstrated their ability to alleviate catastrophic forgetting by continually evolving the model's capacity, thus accommodating the demands of lifelong learning.

**Knowledge Graph-based Lifelong Learning**. methods have emerged as a promising avenue in the field of CL, offering a structured approach to managing and transferring knowledge across tasks. These methods leverage the idea of constructing a knowledge graph that represents the relationships and dependencies between different tasks and their associated information. A notable contribution in this category is the Knowledge Distillation approach by (Li et al., 2022). This approach employs a knowledge graph to model the relationships between tasks and uses this graph to guide the distillation process during learning. Another significant approach is the task-embedded control graph proposed by (Parisotto & Salakhutdinov, 2017), which employs a graphical representation of tasks and their dependencies, enabling efficient task switching and knowledge transfer. Knowledge-graph based methods have shown promise in alleviating catastrophic forgetting by structuring the knowledge learned across tasks and allowing models to retain and utilize previously acquired information effectively.

**Hypernetwork based Lifelong Learning.** The hypernetwork-based CL (Liang et al., 2023) leverages a task-conditioned neural network to generate the parameters $\Theta_t = h(e_t; \Psi)$, where $e_t$ is the trainable task embedding for task $t$ and $h(\cdot; \Psi)$ is a neural network, shared by all tasks and parameterized by the weights $\Psi$. The use of hypernetworks has emerged as a promising avenue in the field of lifelong learning. (Von Oswald et al., 2019) introduced hypernetworks as a solution for mitigating catastrophic forgetting, allowing neural networks to dynamically generate weights. (Brahma et al., 2021) explored hypernetworks as a means to adapt neural architectures to new tasks, thus enhancing adaptability and efficiency in lifelong learning scenarios. Hypernetworks present an exciting prospect for parameter-efficient and adaptable models in continual learning.

**Parameter-efficient Lifelong Learning**. As the demand for parameter-efficient lifelong learning models grows, research has shifted towards resource-constrained environments. Progressive Neural Networks (Rusu et al., 2016) have played a pivotal role in this regard by introducing methods for transferring knowledge across tasks while minimizing the growth of model parameters. Additionally, approaches like Elastic Weight Consolidation (Kirkpatrick et al., 2017) have contributed significantly to the realm of parameter-efficient lifelong learning by safeguarding previously learned knowledge. These techniques address the critical issue of scalability in continual learning settings.

