# OpenReview forum: "Diversity-aware Continual Learning with Latent Knowledge Hypergraph"
_ICLR.cc/2024/Conference — ICLR 2024 Conference Withdrawn Submission_

### Official Review · Reviewer_RNxE · 2023-10-28

**Soundness:** 3 good
**Presentation:** 3 good
**Contribution:** 3 good
**Rating:** 5
**Confidence:** 4

**Summary:**

To address the trade off between parameter efficiency and capacity saturation in the continual learning problem, the HyperGKL method was proposed. The hypergraph was used to model the skill-task relationship, and explicit constrains were applied. Experimental results shew the efficiency of the proposed method.

**Strengths:**

1) The paper is clear and logical.
2) The proposed latent skill-task knowledge hypergraph is effective compared with other group/parameter-level methods.
3) Experimental results show the effectiveness of the proposed method, supporting the statements in the thesis.

**Weaknesses:**

1) Some parts of the explanation are not clear enough, e.g., the third challenge in the last paragraph in section 2, that c is a non-differentiable function. I am confused about how it is solved and what is the explicit definition of c. Maybe it was mentioned in the paper, but I didn't see it. Hopefully, the author will explain it in a more prominent place.
2) As mentioned in section 3.2, the novel knowledge discovery and hypergraph growth are based on the k given by the evolutionary search. So I wonder if this step introduces too much time consumption and reduces training efficiency compared to other comparison methods.

**Questions:**

1) Some parts of the explanation are not clear enough, e.g., the third challenge in the last paragraph in section 2, that c is a non-differentiable function. I am confused about how it is solved and what is the explicit definition of c. Maybe it was mentioned in the paper, but I didn't see it. Hopefully the author will explain it in a more prominent place.
2) As mentioned in section 3.2, the novel knowledge discovery and hypergraph growth are based on the k given by the evolutionary search. So I wonder if this step introduces too much time consumption and reduces training efficiency compared to other comparison methods.
3) Typo: tasj -> task in the first paragraph in section 3.1, networj -> network in the last paragraph in Section 3.3

---

### Official Review · Reviewer_j4Dh · 2023-10-31

**Soundness:** 1 poor
**Presentation:** 3 good
**Contribution:** 2 fair
**Rating:** 3
**Confidence:** 5

**Summary:**

To efficiently increase parameters in proportion to the actual capacity required by each new task, this work proposes a method, HyperGKL. It estimates task diversity based on latent knowledge across tasks to balance parameter efficiency and capacity saturation. The proposed method proposes parameter efficient constraints in CL objective and incorporates designs to enforce these properties into the HyperGKL framework. It demonstrates that proposed HyperGKL outperforms non-rehearsal methods on various small datasets in terms of performance and parameter efficiency. HuperGKL also shows expected knowledge sharing and task diversity behavior in learned latent knowledge hypergraphs.

**Strengths:**

This paper focuses on an overlooked area in continual learning i.e., a trade-off between parameter efficiency and capacity saturation. It proposes a dynamic latent knowledge hypergraph method that aims to find a good trade-off.

It imposes parameter-efficiency constraints defined by task diversity on the CL objective to keep the number of parameters in check. It incorporates these attributes into a HyperNet via trainable embeddings for capturing shared knowledge among tasks and hyperedges for capturing task diversity.

HyperGKL outperforms different non-rehearsal CL methods on various small datasets in terms of performance and parameter efficiency. It also demonstrates expected behavior in learned latent knowledge hypergraphs.

**Weaknesses:**

I am not convinced that capacity saturation (under parameterization) is a prevalent issue in continual learning (CL), given that most modern deep neural networks (DNNs) are sufficiently large or over parameterized to accommodate knowledge from a large sequence of tasks. Many CL methods based on a single DNN (5-10M parameters) achieve competitive performance in large scale memory constrained CL problems e.g., ImageNet classification task. Rather capacity saturation might be a limitation of specific design choices for CL models.

Lack of experiments on large scale problem settings with large datasets e.g., ImageNet (1000 classes, 1.28M images, higher number of tasks) which may reveal if capacity saturation assumption holds. Given the capacity saturation motivation of this work, chosen experimental settings based on small datasets (small number of tasks) may not exhibit capacity saturation and thus may not need dynamic expansion mechanisms to solve these small tasks. Many CL methods work only on these toy datasets and fail for larger datasets. Should test on multiple distributions rather than only ones that are as non-iid as possible. Many baselines are old and extremely weak, e.g., EWC fails to perform well for incremental class learning and for large-scale datasets.

 Also efficacy of CL models on small datasets is not predictive of their efficacy on large datasets. So more challenging settings may be required to verify HyperGKL’s effectiveness.

It is unclear how task sequence is constructed and how data ordering (with or without class boundaries) intensifies catastrophic forgetting. It is unclear how HyperGKL performs in different data orderings e.g., class incremental learning (with class boundaries, severe forgetting) and IID (classes revisit, less forgetting).

Comparisons do not include rehearsal-based or stronger dynamic expansion SOTA CL baselines. It is unclear if HyperGKL provides performance competitive to recent SOTA methods. Given all the complexities in designing HyperGKL, it is worth checking the advantages and disadvantages of this method compared to other simple CL methods with a single DNN.

Computational overhead (number of SGD steps or iterations) and/or training time have not been reported. Besides memory, compute is also critical for real world applicability. Continual learning needs to demonstrate some benefit for real-world applications to really have impact.

Performance improvements over comparison methods are negligible even on these toy datasets.

There is no justification for not testing on largescale datasets. All experiments were done on an A100, and people have been doing continual learning on larger datasets like ImageNet on far weaker GPUs over the past 7 year or so.

**Questions:**

Does inclusion of parameter-efficiency constraints in the proposed CL objective increase computational cost?

Does your method require task labels during inference or test time?

Although the proposed framework presents valuable insights, considering all intricacies in designing proposed HyperGKL and training it, why do people choose HyperGKL over other simple CL methods e.g., rehearsal methods with a single DNN?

In settings where the CL model has to adapt to a large sequence of tasks, does the proposed framework increase computational overhead and training time?

How do you set the number of nodes and hyperedges given a particular task?

How do you measure capacity saturation? How is capacity saturation relevant to chosen experimental settings based on a small number of tasks?

How does HyperGKL generalize to different problem settings?

---

### Official Review · Reviewer_9KLn · 2023-11-01

**Soundness:** 3 good
**Presentation:** 2 fair
**Contribution:** 2 fair
**Rating:** 5
**Confidence:** 4

**Summary:**

The paper proposes a method to strike a balance between the capacity saturation and parameter efficiency in online task-incremental continual learning setting. The proposed method involves using a hypergraph structure to quantify the task diversity and estimate the necessary capacity for the new tasks. Evaluation is done on four continual learning datasets, where the proposed method shows the highest accuracy amongst compared arts.

**Strengths:**

- The trade-off between capacity saturation and parameter efficiency is an interesting point that is not discussed frequently
- The idea of modeling a task diversity using hyper graphs seem novel

**Weaknesses:**

- The setting in which capacity saturation and parameter efficiency really matters is, as mentioned in the manuscript, when CL methods are deployed on edge devices with little memory. This is the main reason why replay-based CL methods are seldom compared to in the submission but I disagree. If using a replay buffer is too costly, what's to say that expanding the model size (however efficiently that might be) is not too costly? Replay based CL methods can be understood as having the best parameter efficiency (as the model size does not grow) and tackling the capacity saturation issue with past task samples in the replay buffer. I think more comparisons both in terms of concepts and experiments should be done against replay based CL methods to really justify the proposed trade-off in the paper.

**Questions:**

Please see the weaknesses.

---

### Official Review · Reviewer_8ANK · 2023-11-02

**Soundness:** 2 fair
**Presentation:** 1 poor
**Contribution:** 2 fair
**Rating:** 3
**Confidence:** 3

**Summary:**

The paper places itself in the field of continual learning (CL), specifically task-incremental continual learning where new tasks arrive sequentially with an assumed simultaneous input and label distribution shift.
The paper highlights a solution to a specific problem in this domain: the tradeoff between parameter efficiency and capacity saturation, which as I understand amounts to finding a more optimal model capacity to be able to generalise to all tasks while not having too many parameters.
The solution proposed for this specific problem revolves around the "skill-task knowledge hypergraph" representation that aims to separate and consolidate the different "skills" needed to achieve good accuracy on different classification tasks appearing sequentially in the training.

**Strengths:**

I have found several strengths to this paper:
- The quest to find the right model capacity that balances between generalisation power and compactness is an interesting question to address in a continual learning setting involving several sequential tasks, which this paper tackles
- The experiments are run 5 times with different random seeds (I assume the results are the average), which is sound
- Figure 1 is very clear and helps conveying the method goal

**Weaknesses:**

Content:
The results on accuracy are marginally better, but the results on parameter size do not seem to support the claim that the method achieves better balancing between generalisation power and parameter compactness.

Form:
Overall, I found the presentation of the paper pretty poor. The formatting lacks consistency and restrain, needlessly convoluted language sometimes hides the point, and the work isn't contextualised enough as the related works section is put in the supplementary material, which makes the baselines section seem very ad hoc. Putting the results on half the datasets
Here are examples of specific concerns that highlight my impression on the presentation:

- Too much bold, italic, bold+italic and even underlined formatting that isn't always necessary and actually distracts from reading, I understand this might be intended to help the reader skimming over the paper but it achieves the opposite
- References section not cleaned up as it contains duplicates
- The results on half the datasets are put in the supplementary material
- Explanation of eq 6 lacks clarity
- Unclear syntax and typos (I didn't list all of them but give some examples):
    - after eq1: without the access of -> without access to?
    - to summary -> to summarise
    - "til" isn't a proper word
    - "without the access of T1, T2..." -> without access to
    - typos are normal, english isn't my first language so I understand, but a publication should be proofread to avoid these

**Questions:**

- The function c defined after eq 2 is said to measure the "size" of gradients / weights, and I am assuming it is related to the actual memory footprint. Is this assumption correct? To achieve the goal of reducing this size, the method describes among other things a step designed to find the optimal number of new skills, using an evolutionary search. How does this search to reduce space complexity impacts time complexity?
- Results on TPS and IPS do not seem better than most other methods especially regularization-based approaches. At best it seems on par. Am I missing something?